# SIMPLE GNN REGULARISATION FOR 3D MOLECULAR PROPERTY PREDICTION & BEYOND

**Jonathan Godwin, Michael Schaarschmidt, Alexander Gaunt,**
**Alvaro Sanchez-Gonzales, Yulia Rubanova, Petar Veličković,**
**James Kirkpatrick & Peter Battaglia**
DeepMind, London
`{jonathangodwin}@deepmind.com`

## ABSTRACT

In this paper we show that simple noisy regularisation can be an effective way to address oversmoothing. We argue that regularisers addressing oversmoothing should both penalise node latent similarity and encourage meaningful node representations. From this observation we derive "Noisy Nodes", a simple technique in which we corrupt the input graph with noise, and add a noise correcting node-level loss. The diverse node level loss encourages latent node diversity, and the denoising objective encourages graph manifold learning. Our regulariser applies well-studied methods in simple, straightforward ways which allow even generic architectures to overcome oversmoothing and achieve state of the art results on quantum chemistry tasks, and improve results significantly on Open Graph Benchmark (OGB) datasets. Our results suggest Noisy Nodes can serve as a complementary building block in the GNN toolkit.

## 1 INTRODUCTION

Graph Neural Networks (GNNs) are a family of neural networks that operate on graph structured data by iteratively passing learned messages over the graph's structure (Scarselli et al., 2009; Bronstein et al., 2017; Gilmer et al., 2017; Battaglia et al., 2018; Shlomi et al., 2021). While Graph Neural Networks have demonstrated success in a wide variety of tasks (Zhou et al., 2020a; Wu et al., 2020; Bapst et al., 2020; Schütt et al., 2017; Klicpera et al., 2020a), it has been proposed that in practice "oversmoothing" limits their ability to benefit from overparametrization.

Oversmoothing is a phenomenon where a GNN's latent node representations become increasing indistinguishable over successive steps of message passing (Chen et al., 2019). Once these representations are oversmoothed, the relational structure of the representation is lost, and further message-passing cannot improve expressive capacity. We argue that the challenges of overcoming oversmoothing are two fold. First, finding a way to encourage node latent diversity; second, to encourage the diverse node latents to encode meaningful graph representations. Here we propose a simple noise regulariser, Noisy Nodes, and demonstrate how it overcomes these challenges across a range of datasets and architectures, achieving top results on OC20 IS2RS & IS2RE direct, QM9 and OGBG-PCQM4Mv1.

Our "Noisy Nodes" method is a simple technique for regularising GNNs and associated training procedures. During training, our noise regularisation approach corrupts the input graph's attributes with noise, and adds a per-node noise correction term. We posit that our Noisy Nodes approach is effective because the model is rewarded for maintaining and refining distinct node representations through message passing to the final output, which causes it to resist oversmoothing. Like denoising autoencoders, it encourages the model to explicitly learn the manifold on which the uncorrupted input graph's features lie, analogous to a form of representation learning. When applied to 3D molecular prediction tasks, it encourages the model to distinguish between low and high energy states. We find that applying Noisy Nodes reduces oversmoothing for shallower networks, and allows us to see improvements with added depth, even on tasks for which depth was assumed to be unhelpful.

This study's approach is to investigate the combination of Noisy Nodes with generic, popular baseline GNN architectures. For 3D Molecular prediction we use a standard architecture working on 3D point clouds developed for particle fluid simulations, the Graph Net Simulator (GNS) (Sanchez-Gonzalez*

et al., 2020), which has also been used for molecular property prediction (Hu et al., 2021b). Without using Noisy Nodes the GNS is not a competitive model, but using Noisy Nodes allows the GNS to achieve top performance on three 3D molecular property prediction tasks: the OC20 IS2RE direct task by 43% over previous work, 12% on OC20 IS2RS direct, and top results on 3 out of 12 of the QM9 tasks. For non-spatial GNN benchmarks we test a MPNN (Gilmer et al., 2017) on OGBG-MOLPCBA and OGBG-PCQM4M (Hu et al., 2021a) and again see significant improvements. Finally, we applied Noisy Nodes to a GCN (Kipf & Welling, 2016), arguably the most popular and simple GNN, trained on OGBN-Arxiv and see similar results. These results suggest Noisy Nodes can serve as a complementary GNN building block.

## 2 PRELIMINARIES: GRAPH PREDICTION PROBLEM

Let $G = (V, E, g)$ be an input graph. The nodes are $V = \{v_1, \ldots, v_{|V|}\}$, where $v_i \in \mathbb{R}^{d_v}$. The directed, attributed edges are $E = \{e_1, \ldots, e_{|E|}\}$: each edge includes a sender node index, receiver node index, and edge attribute, $e_k = (s_k, r_k, e_k)$, respectively, where $s_k, r_k \in \{1, \ldots, |V|\}$ and $e_k \in \mathbb{R}^{d_e}$. The graph-level property is $g \in \mathbb{R}^{d_g}$.

The goal is to predict a target graph, $G'$, with the same structure as $G$, but different node, edge, and/or graph-level attributes. We denote $\hat{G}'$ as a model's prediction of $G'$. Some error metric defines quality of $\hat{G}'$ with respect to the target $G'$, $\text{Error}(\hat{G}', G')$, which the training loss terms are defined to optimize. In this paper the phrase "message passing steps" is synonymous with "GNN layers".

## 3 OVERSMOOTHING

"Oversmoothing" is when the node latent vectors of a GNN become very similar after successive layers of message passing. Once nodes are identical there is no relational information contained in the nodes, and no higher-order latent graph representations can be learned. It is easiest to see this effect with the update function of a Graph Convolutional Network with no adjacency normalization $v_i^k = \sum_j W v_j^{k-1}$ with $j \in Neighborhood_{v_i}, W \in \mathbb{R}^{d_g \times d_g}$ and $k$ the layer index. As the number of applications increases, the averaging effect of the summation forces the nodes to become almost identical. However, as soon as residual connections are added we can construct a network that need not suffer from oversmoothing by setting the residual updates to zero at a similarity threshold. Similarly, multi-head attention Vaswani et al. (2017); Veličković et al. (2018) and GNNs with edge updates (Battaglia et al., 2018; Gilmer et al., 2017) can modulate node updates. As such for modern GNNs oversmoothing is primarily a "training" problem - i.e. how to choose model architectures and regularisers to encourage and preserve meaningful latent relational representations.

We can discern two desiderata for a regulariser or loss that addresses oversmoothing. First, it should penalise identical node latents. Second, it should encourage meaningful latent representations of the data. One such example may be the auto-regressive loss of transformer based language models (Brown et al. (2020)). In this case, each word (equivalent to node) prediction must be distinct, and the auto-regressive loss encourages relational dependence upon prior words. We can take inspiration from this observation to derive auxiliary losses that both have diverse node targets and encourage relational representation learning. In the following section we derive one such regulariser, Noisy Nodes.

## 4 NOISY NODES

Noisy Nodes tackles the oversmoothing problem by adding a diverse noise correction target, modifying the original graph prediction problem definition in several ways. It introduces a graph corrupted by noise, $\tilde{G} = (\tilde{V}, \tilde{E}, \tilde{g})$, where $\tilde{v}_i \in \tilde{V}$ is constructed by adding noise, $\sigma_i$, to the input nodes, $\tilde{v}_i = v_i + \sigma_i$. The edges, $\tilde{E}$, and graph-level attribute, $\tilde{g}$, can either be uncorrupted by noise (i.e., $\tilde{E} = E, \tilde{g} = g$), calculated from the noisy nodes (for example in a nearest neighbors graph), or corrupted independent of the nodes—these are minor choices that can be informed by the specific problem setting.

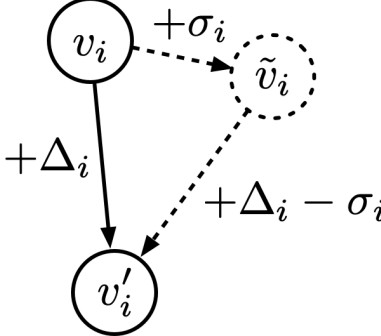
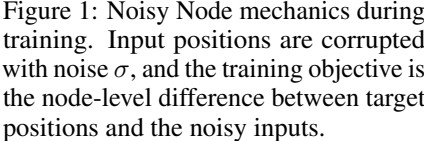

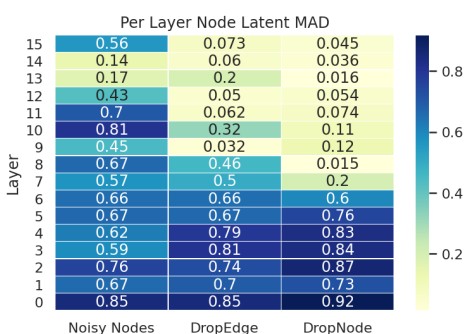

Figure 1: Noisy Node mechanics during training. Input positions are corrupted with noise $\sigma$, and the training objective is the node-level difference between target positions and the noisy inputs.

Figure 2: Per layer node latent diversity, measured by MAD on a 16 layer MPNN trained on OGBG-MOLPCBA. Noisy Nodes maintains a higher level of diversity throughout the network than competing methods.

Our method requires a noise correction target to prevent oversmoothing by enforcing diversity in the last layers of the GNN, which can be achieved with an auxiliary denoising autoencoder loss. For example, where the Error is defined with respect to graph-level predictions (e.g., predict the minimum energy value of some molecular system), a second output head can be added to the GNN architecture which requires denoising the inputs as targets. Alternatively, if the inputs and targets are in the same real domain as is the case for physical simulations we adjust the target for the noise. Figure 1 demonstrates this Noisy Nodes set up. The auxiliary loss is weighted by a constant coefficient $\lambda \in \mathbb{R}$.

In Figure 2 we illustrate the impact of Noisy Nodes on oversmoothing by plotting the Mean Absolute Distance (MAD) (Chen et al., 2020) of the residual updates of each layer of an MPNN trained on the QM9 (Ramakrishnan et al., 2014) dataset, and compare it to alternative methods DropEdge (Rong et al., 2019) and DropNode (Do et al., 2021). MAD is a measure of the diversity of graph node features, often used to quantify oversmoothing, the higher the number the more diverse the node features, the lower the number the less diverse. In this plot we can see that for Noisy Nodes the node updates remain diverse for all of the layers, whereas without Noisy Nodes diversity is lost. Further analysis of MAD across seeds and with sorted layers can be seen in Appendix Figures 7 and 6 for models applied to 3D point clouds.

**The Graph Manifold Learning Perspective.** By using an implicit mapping from corrupted data to clean data, the Noisy Nodes objective encourages the model to learn the manifold on which the clean data lies— we speculate that the GNN learns to go from low probability graphs to high probability graphs. In the autoencoder case the GNN learns the manifold of the input data. When node targets are provided, the GNN learns the manifold of the target data (e.g. the manifold of atoms at equilibrium). We speculate that such a manifold may include commonly repeated substructures that are useful for downstream prediction tasks. A similar motivation can be found for denoising in (Vincent et al., 2010; Song & Ermon, 2019).

**The Energy Perspective for Molecular Property Prediction.** Local, random distortions of the geometry of a molecule at a local energy minimum are almost certainly higher energy configurations. As such, a task that maps from a noised molecule to a local energy minimum is learning a mapping from high energy to low energy. Data such as QM9 contains molecules at local minima.

Some problems have input data that is already high energy, and targets that are at equilibrium. For these datasets we can generate new high energy states by adding noise to the inputs but keeping the equilibrium target the same, Figure 1 demonstrates this approach. To preserve translation invariance we use displacements between input and target $\Delta$, the corrected target after noise is $\Delta - \sigma$.

## 5 RELATED WORK

**Oversmoothing.** Recent work has aimed to understand why it is challenging to realise the benefits of training deeper GNNs (Wu et al., 2020). Since first being noted in ((Li et al., 2018)) oversmoothing has been studied extensively and regularisation techniques have been suggested to overcome it (Chen

et al., 2019; Cai & Wang, 2020; Rong et al., 2019; Zhou et al., 2020b; Yang et al., 2020; Do et al., 2021; Zhao & Akoglu, 2020). A recent paper, (Li et al., 2021), finds, as in previous work, (Li et al., 2019; 2020), the optimal depth for some datasets they evaluate on to be far lower (5 for OGBN-Arxiv from the Open Graph Benchmark (Hu et al., 2020a), for example) than the 1000 layers possible.

**Denoising & Noise Models.** Training neural networks with noise has a long history (Sietsma & Dow, 1991; Bishop, 1995). Of particular relevance are Denoising Autoencoders (Vincent et al., 2008) in which an autoencoder is trained to map corrupted inputs $\tilde{\mathbf{x}}$ to uncorrupted inputs $\mathbf{x}$. Denoising Autoencoders have found particular success as a form of pre-training for representation learning (Vincent et al., 2010). More recently, in research applying GNNs to simulation (Sanchez-Gonzalez et al., 2018; Sanchez-Gonzalez* et al., 2020; Pfaff et al., 2020) Gaussian noise is added during training to input positions of a ground truth simulator to mimic the distribution of errors of the learned simulator. Pre-training methods (Devlin et al., 2019; You et al., 2020; Thakoor et al., 2021) are another similar approach; most similarly to our method Hu et al. (2020b) apply a reconstruction loss to graphs with masked nodes to generate graph embeddings for use in downstream tasks. FLAG (Kong et al., 2020) adds adversarial noise during training to input node features as a form of data augmentation for GNNs that demonstrates improved performance for many tasks. It does not add an additional auxiliary loss, which we find is essential for addressing oversmoothing. In other related GNN work, (Sato et al., 2021) use random input features to improve generalisation of graph neaural networks. Adding noise to help input node disambiguation has also been covered in (Dasoulas et al., 2019; Loukas, 2020; Vignac et al., 2020; Murphy et al., 2019), but there is no auxiliary loss.

Finally, we take inspiration from (Vincent et al., 2008; 2010; Vincent, 2011; Song & Ermon, 2019) which use the observation that noised data lies off the data manifold for representation learning and generative modelling.

**Machine Learning for 3D Molecular Property Prediction.** One application of GNNs is to speed up quantum chemistry calculations which operate on 3D positions of a molecule (Duvenaud et al., 2015; Gilmer et al., 2017; Schütt et al., 2017; Hu et al., 2021b). Common goals are the prediction of molecular properties (Ramakrishnan et al., 2014), forces (Chmiela et al., 2017), energies (Chanussot* et al., 2020) and charges (Unke & Meuwly, 2019).

A common approach to embed physical symmetries is to design a network that predicts a rotation and translation invariant energy (Schütt et al., 2017; Klicpera et al., 2020a; Liu et al., 2021). The input features of such models include distances (Schütt et al., 2017), angles (Klicpera et al., 2020b;a) or torsions and higher order terms (Liu et al., 2021). An alternative approach to embedding symmetries is to design a rotation equivariant neural network that use equivariant representations (Thomas et al., 2018; Köhler et al., 2019; Kondor et al., 2018; Fuchs et al., 2020; Batzner et al., 2021; Anderson et al., 2019; Satorras et al., 2021).

**Machine Learning for Bond and Atom Molecular Graphs**. Predicting properties from molecular graphs without 3D points, such as graphs of bonds and atoms, is studied separately and often used to benchmark generic graph property prediction models such as GCNs (Hu et al., 2020a) or GATs (Veličković et al., 2018). Models developed for 3D molecular property prediction cannot be applied to bond and atom graphs. Common datasets that contain such data are OGBG-MOLPCBA and OGBG-MOLHIV.

## 6 3D Molecular Property Prediction Experiments and Results

In this section we evaluate how a popular, simple model, the GNS (Sanchez-Gonzalez* et al., 2020) performs on 3D molecular prediction tasks when combined with Noisy Nodes. The GNS was originally developed for particle fluid simulations, but has recently been adapted for molecular property prediction (Hu et al., 2021b). We find that Without Noisy Nodes the GNS architecture is not competitive, but by using Noisy Nodes we see improved performance comparable to the use of specialised architectures.

We made minor changes to the GNS architecture. We featurise the distance input features using radial basis functions. We group layer weights, similar to grouped layers used in Jumper et al. (2021) for reduced parameter counts; for a group size of $n$ the first $n$ layer weights are repeated, i.e. the first layer with a group size of 10 has the same weights as the $11^{th}$, $21^{st}$, $31^{st}$ layers and so on. $n$ contiguous

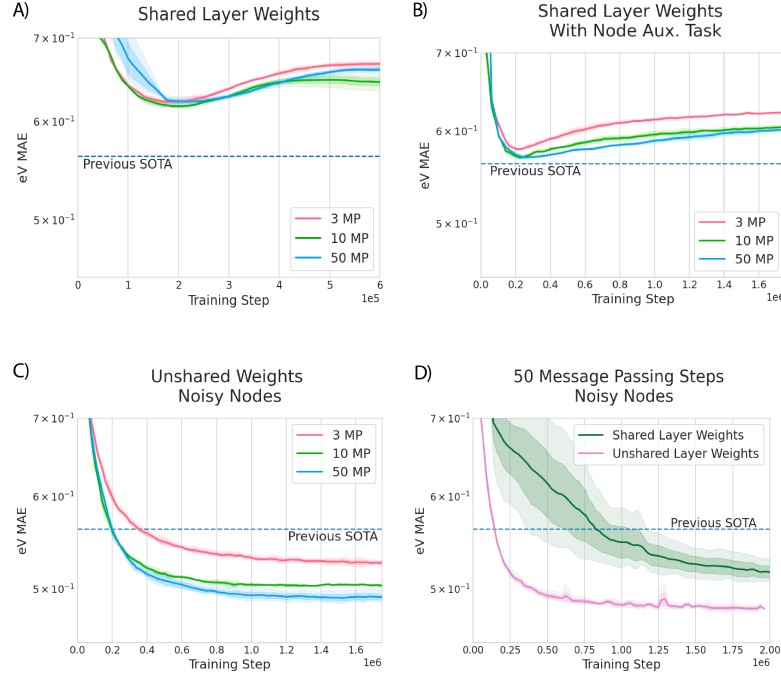

Figure 3: Validation curves, OC20 IS2RE ID. **A)** Without any node targets our model has poor performance and realises no benefit from depth. **B)** After adding a position node loss, performance improves as depth increases. **C)** As we add Noisy Nodes and parameters the model achieves SOTA, even with 3 layers, and stops overfitting. **D)** Adding Noisy Nodes allows a model with even fully shared weights to achieve SOTA.

blocks of layers are considered a single group. Finally we find that decoding the intermediate latents and adding a loss after each group aids training stability. The decoder is shared across groups.

We tested this architecture on three challenging molecular property prediction benchmarks: OC20 (Chanussot* et al., 2020) IS2RS & IS2RE, and QM9 (Ramakrishnan et al., 2014). These benchmarks are detailed below, but as general distinctions, OC20 tasks use graphs 2-20x larger than QM9. While QM9 always requires graph-level prediction, one of OC20's two tasks (IS2RS) requires node-level predictions while the other (IS2RE) requires graph-level predictions. All training details may be found in the Appendix.

## 6.1 Open Catalyst 2020

**Dataset.** The OC20 dataset (Chanussot* et al., 2020) (CC Attribution 4.0) describes the interaction of a small molecule (the adsorbate) and a large slab (the catalyst), with total systems consisting of 20-200 atoms simulated until equilibrium is reached.

We focus on two tasks; the Initial Structure to Resulting Energy (IS2RE) task which takes the initial structure of the simulation and predicts the final energy, and the Initial Structure to Resulting Structure (IS2RS) which takes the initial structure and predicts the relaxed structure. Note that we train the more common "direct" prediction task that map directly from initial positions to target in a single forward pass, and compare against other models trained for direct prediction.

Models are evaluated on 4 held out test sets. Four canonical validation datasets are also provided. Test sets are evaluated on a remote server hosted by the dataset authors with a very limited number of submissions per team.

Noisy Nodes in this case consists of a random jump between the initial position and relaxed position. During training we first sample uniformly from a point in the relaxation trajectory or interpolate uniformly between the initial and final positions $(v_i - \tilde{v}_i)\gamma, \gamma \sim U(0,1)$, and then add I.I.D Gaussian noise with mean zero and $\sigma = 0.3$. The Noisy Node target is the relaxed structure.

Table 1: OC20 ISRE Validation, eV MAE, $\downarrow$.
"GNS-Shared" indicates shared weights. "GNS-10" indicates a group size of 10.

| Model | Layers | OOD Both | OOD Adsorbate | OOD Catalyst | ID |
|---|---|---|---|---|---|
| GNS | 50 | 0.59 $\pm$0.01 | 0.65 $\pm$0.01 | 0.55 $\pm$0.00 | 0.54 $\pm$0.00 |
| GNS-Shared + Noisy Nodes | 50 | 0.49 $\pm$0.00 | 0.54 $\pm$0.00 | 0.51 $\pm$0.01 | 0.51 $\pm$0.01 |
| GNS + Noisy Nodes | 50 | 0.48 $\pm$0.00 | 0.53 $\pm$0.00 | 0.49 $\pm$0.01 | 0.48 $\pm$0.00 |
| GNS-10 + Noisy Nodes | 100 | **0.46**$\pm$0.00 | **0.51** $\pm$0.00 | **0.48** $\pm$0.00 | **0.47** $\pm$0.00 |

Table 2: Results OC20 IS2RE Test

| eV MAE $\downarrow$ | | | | |
|---|---|---|---|---|
| | SchNet | DimeNet++ | SpinConv | SphereNet | GNS + Noisy Nodes |
| OOD Both | 0.704 | 0.661 | 0.674 | 0.638 | **0.465 (-24.0%)** |
| OOD Adsorbate | 0.734 | 0.725 | 0.723 | 0.703 | **0.565 (-22.8%)** |
| OOD Catalyst | 0.662 | 0.576 | 0.569 | 0.571 | **0.437 (-17.2%)** |
| ID | 0.639 | 0.562 | 0.558 | 0.563 | **0.422 (-18.8%)** |
| Average Energy within Threshold (AEwT) $\uparrow$ | | | | |
| | SchNet | DimeNet++ | SpinConv | SphereNet | GNS + Noisy Nodes |
| OOD Both | 0.0221 | 0.0241 | 0.0233 | 0.0241 | **0.047 (+95.8%)** |
| OOD Adsorbate | 0.0233 | 0.0207 | 0.026 | 0.0229 | **0.035 (+89.5%)** |
| OOD Catalyst | 0.0294 | 0.0410 | 0.0382 | 0.0409 | **0.080 (+95.1%)** |
| ID | 0.0296 | 0.0425 | 0.0408 | 0.0447 | **0.091 (+102.0%)** |

We first convert to fractional coordinates (i.e. use the periodic unit cell as the basis) which render the predictions of our model invariant to rotations, and append the following rotation and translation invariant vector $(\alpha\beta^T, \beta\gamma^T, \alpha\gamma^T, |\alpha|, |\beta|, |\gamma|) \in \mathbb{R}^6$ to the edge features where $\alpha, \beta, \gamma$ are vectors of the unit cell. This additional vector provides rotation invariant angular and extent information to the GNN.

**IS2RE Results.** In Figure 3 we show how using Noisy Nodes allows the GNS to achieve state of the art performance. Figure 3 A shows that without any auxiliary node target, an IS2RE GNS achieves poor performance even with increased depth. The fact that increased depth does not result in improvement supports the hypothesis that GNS suffers from oversmoothing. As we add a node level position target in B) we see better performance, and improvement as depth increases, validating our hypothesis that node level targets are key to addressing oversmoothing. In C) we add noisy nodes and parameters, and see that the increased diversity of the node level predictions leads to very significant improvements and SOTA, even for a shallow 3 layer network. D) demonstrates this effect is not just due to increased parameters - SOTA can still be achieve with shared layer weights .

In Table 1 we conduct an ablation on our hyperparameters, and again demonstrate the improved performance of using Noisy Nodes. Results were averaged over 3 seeds and standard errors on the best obtained checkpoint show little sensitivity to initialisation. All results in the table are reported using sampling states from trajectories. We conducted an ablation on ID comparing sampling from a relaxation trajectory and interpolating between initial & final positions which found that interpolation improved our score from 0.47 to 0.45.

Our best hyperparameter setting was 100 layers which achieved a 95.6% relative performance improvement against SOTA results (Table 2) on the AEwT benchmark. Due to limited permitted test submissions, results presented here were from one test upload of our best performing validation seed.

**IS2RS Results.** In Table 4 we see that GNS + Noisy Nodes is significantly better than the only other reported IS2RS direct result, ForceNet, itself a GNS variant.

Table 3: OC20 IS2RS Validation, ADwT, ↑

| Model | Layers | OOD Both | OOD Adsorbate | OOD Catalyst | ID |
|-------|--------|----------|---------------|--------------|-----|
| GNS | 50 | 43.0%±0.0 | 38.0%±0.0 | 37.5% 0.0 | 40.0%±0.0 |
| GNS + Noisy Nodes | 50 | 50.1%±0.0 | 44.3%±0.0 | 44.1%±0.0 | 46.1% ±0.0 |
| GNS-10 + Noisy Nodes | 50 | 52.0%±0.0 | 46.2%±0.0 | 46.1% ±0.0 | 48.3% ±0.0 |
| GNS-10 + Noisy Nodes + Pos only | 100 | **54.3%**±0.0 | **48.3%**±0.0 | **48.2%** ±0.0 | **50.0%** ±0.0 |

Table 4: OC20 IS2RS Test, ADwT, ↑

| Model | OOD Both | OOD Adsorbate | OOD Catalyst | ID |
|-------|----------|---------------|--------------|-----|
| ForceNet | 46.9% | 37.7% | 43.7% | 44.9% |
| GNS + Noisy Nodes | **52.7%** | **43.9%** | **48.4%** | **50.9%** |
| Relative Improvement | **+12.4%** | **+16.4%** | **+10.7%** | **+13.3%** |

## 6.2 QM9

**Dataset.** The QM9 benchmark (Ramakrishnan et al., 2014) contains 134k molecules in equilibrium with up to 9 heavy C, O, N and F atoms, targeting 12 associated chemical properties (License: CCBY 4.0). We use 114k molecules for training, 10k for validation and 10k for test. All results are on the test set. We subtract a fixed per atom energy from the target values computed from linear regression to reduce variance. We perform training in eV units for energetic targets, and evaluate using MAE. We summarise the results across the targets using mean standardised MAE (std. MAE) in which MAEs are normalised by their standard deviation, and mean standardised logMAE. Std. MAE is dominated by targets with high relative error such as $\Delta\epsilon$, whereas logMAE is sensitive to outliers such as $\langle R^2 \rangle$. As is standard for this dataset, a model is trained separately for each target.

For this dataset we add I.I.D Gaussian noise with mean zero and $\sigma = 0.02$ to the input atom positions. A denoising autoencoder loss is used.

**Results** In Table 6 we can see that adding Noisy Nodes significantly improves results by 23.1% relative for GNS, making it competitive with specialised architectures. To understand the effect of adding a denoising loss, we tried just adding noise and found no where near the same improvement (Table 6).

A GNS-10 + Noisy Nodes with 30 layers achieves top results on 3 of the 12 targets and comparable performance on the remainder (Table 6). On the std. MAE aggregate metric GNS + Noisy Nodes performs better than all other reported results, showing that Noisy Nodes can make even a generic model competitive with models hand-crafted for molecular property prediction. The same trend is repeated for an rotation invariant version of this network that uses the principle axes of inertia ordered by eigenvalue as the co-ordinate frame (Table 5).

$\langle R^2 \rangle$, the electronic spatial extent, is an outlier for GNS + Noisy Nodes. Interestingly, we found that without noise GNS-10 + Noisy Nodes achieves 0.33 for this target. We speculate that this target is particularly sensitive to noise, and the best noise value for this target would be significantly lower than for the dataset as a whole.

Table 5: QM9, Impact of Noisy Nodes on GNS architecture.

| | Layers | std. MAE | % Change | logMAE |
|-------|--------|----------|----------|--------|
| GNS | 10 | 1.17 | - | -5.39 |
| GNS + Noise But No Node Target | 10 | 1.16 | -0.9% | -5.32 |
| GNS + Noisy Nodes | 10 | 0.90 | -23.1% | -5.58 |
| GNS-10 + Noisy Nodes | 20 | 0.89 | -23.9% | -5.59 |
| GNS-10 + Noisy Nodes + Invariance | 30 | 0.92 | -21.4% | -5.57 |
| GNS-10 + Noisy Nodes | 30 | **0.88** | **-24.8%** | **-5.60** |

Table 6: QM9, Test MAE, Mean & Standard Deviation of 3 Seeds Reported.

| Target | Unit | SchNet | E(n)GNN | DimeNet++ | SphereNet | PaiNN | **GNS + Noisy Nodes** |
|---|---|---|---|---|---|---|---|
| $\mu$ | D | 0.033 | 0.029 | 0.030 | 0.027 | **0.012** | 0.025 $\pm$0.01 |
| $\alpha$ | $a_0{}^3$ | 0.235 | 0.071 | **0.043** | 0.047 | 0.045 | 0.052 $\pm$0.00 |
| $\epsilon_{\text{HOMO}}$ | meV | 41 | 29.0 | 24.6 | 23.6 | 27.6 | **20.4** $\pm$0.2 |
| $\epsilon_{\text{LUMO}}$ | meV | 34 | 25.0 | 19.5 | 18.9 | 20.4 | **18.6** $\pm$0.4 |
| $\Delta\epsilon$ | meV | 63 | 48.0 | 32.6 | 32.3 | 45.7 | **28.6** $\pm$0.1 |
| $\langle R^2 \rangle$ | $a_0{}^2$ | **0.07** | 0.11 | 0.33 | 0.29 | 0.07 | 0.70 $\pm$0.01 |
| ZPVE | meV | 1.7 | 1.55 | 1.21 | **1.12** | 1.28 | 1.16 $\pm$0.01 |
| $U_0$ | meV | 14.00 | 11.00 | 6.32 | 6.26 | **5.85** | 7.30 $\pm$0.12 |
| $U$ | meV | 19.00 | 12.00 | 6.28 | 7.33 | **5.83** | 7.57 $\pm$0.03 |
| $H$ | meV | 14.00 | 12.00 | 6.53 | 6.40 | **5.98** | 7.43$\pm$0.06 |
| $G$ | meV | 14.00 | 12.00 | 7.56 | 8.0 | **7.35** | 8.30 $\pm$0.14 |
| $c_{\text{v}}$ | $\frac{\text{cal}}{\text{mol K}}$ | 0.033 | 0.031 | 0.023 | **0.022** | 0.024 | 0.025 $\pm$0.00 |
| std. MAE | % | 1.76 | 1.22 | 0.98 | 0.94 | 1.00 | **0.88** |
| logMAE | | -5.17 | -5.43 | -5.67 | -5.68 | **-5.85** | -5.60 |

Table 7: OGBG-PCQM4M Results

| Model | Number of Layers | Using Noisy Nodes | MAE |
|---|---|---|---|
| MPNN + Virtual Node | 16 | Yes | 0.1249 $\pm$ 0.0003 |
| MPNN + Virtual Node | 50 | No | 0.1236 $\pm$ 0.0001 |
| Graphormer (Ying et al., 2021) | - | - | 0.1234 |
| MPNN + Virtual Node | 50 | Yes | **0.1218 $\pm$ 0.0001** |

# 7 NON-SPATIAL TASKS

The previous experiments use the 3D geometries of atoms, and models that operate on 3D points. However, the recipe of adding a denoising auxiliary loss can be applied to other graphs with different types of features. In this section we apply Noisy Nodes to additional datasets with no 3D points, using different GNNs, and show analagous effects to the 3D case. Details of the hyperparameters, models and training details can be found in the appendix.

## 7.1 OGBG-PCQM4M

This dataset from the OGB benchmarks consists of molecular graphs which consist of bonds and atom types, and no 3D or 2D coordinates. To adapt Noisy Nodes to this setting, we randomly flip node and edge features at a rate of 5% and add a reconstruction loss. We evaluate Noisy Nodes using an MPNN + Virtual Node (Gilmer et al., 2017). The test set is not currently available for this dataset.

In Table 7 we see that for this task Noisy Nodes enables a 50 layer MPNN to reach state of the art results. Before adding Noisy Nodes, adding capacity beyond 16 layers did not improve results.

## 7.2 OGBG-MOLPCBA

The OGBG-MOLPCBA dataset contains molecular graphs with no 3D points, with the goal of classifying 128 biological activities. On the OGBG-MOLPCBA dataset we again use an MPNN + Virtual Node and random flipping noise. In Figure 4 we see that adding Noisy Nodes improves the performance of the base model, accentuated for deeper networks. Our 16 layer MPNN improved from 27.6% $\pm$ 0.004 to 28.1% $\pm$ 0.002 Mean Average Precision ("Mean AP"). Figure 5 demonstrates how Noisy Nodes improves performance during training. Of the reported results, our MPNN is most similar to GCN[1] + Virtual Node and GIN + Virtual Node (Xu et al., 2018) which report results of 24.2% $\pm$ 0.003 and 27.03% $\pm$ 0.003 respectively. We evaluate alternative methods for

---

[1]The GCN implemented in the official OGB code base has explicit edge updates, akin to the MPNN.

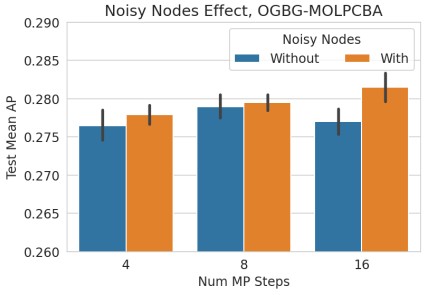

Figure 4: Adding Noisy Nodes with random flipping of input categories improves the performance of MPNNs, and the effect is accentuated with depth.

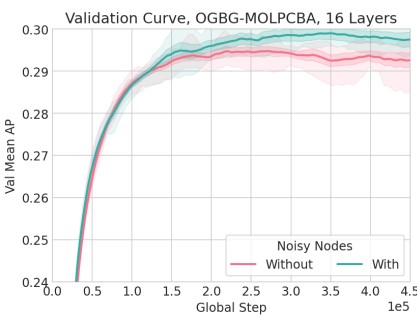

Figure 5: Validation curve comparing with and without noisy nodes. Using Noisy Nodes leads to a consistent improvement.

oversmoothing, DropNode and DropEdge in Figure 2 and find that Noisy Nodes is more effective at address oversmoothing, although all 3 methods can be combined favourably (results in appendix).

### 7.3 OGBN-ARXIV

The above results use models with explicit edge updates, and are reported for graph prediction. To test the effectiveness with Noisy Nodes with GCNs, arguably the simplest and most popular GNN, we use OGBN-ARXIV, a citation network with the goal of predicting the arxiv category of each paper. Adding Noisy Nodes, with noise as input dropout of 0.1, to 4 layer GCN with residual connections improves from 72.39% ± 0.002 accuracy to 72.52% ± 0.003 accuracy. A baseline 4 layer GCN on this dataset reports 71.71% ± 0.002. The SOTA for this dataset is 74.31% (Sun & Wu, 2020).

### 7.4 LIMITATIONS

We have not demonstrated the effectiveness of Noisy Nodes in small data regimes, which may be important for learning from experimental data. The representation learning perspective requires access to a local minimum configuration, which is not the case for all quantum modeling datasets. We have also not demonstrated the combination of Noisy Nodes with more sophisticated 3D molecular property prediction models such as DimeNet++(Klicpera et al., 2020a), such models may require an alternative reconstruction loss to position change, such as pairwise interatomic distances. We leave this to future work.

Noisy Nodes requires careful selection of the form of noise, and a balance between the auxiliary and primary losses. This can require hyper parameter tuning, and models can be sensitive to the choice of these parameters. Noisy Nodes has a particular effect for deep GNNs, but depth is not always an advantage. There are situations, for example molecular dynamics, which place a premium on very fast inference time. However even at 3 layers (a comparable depth to alternative architectures) the GNS architecture achieves state of the art validation OC20 IS2RE predictions (Figure 3). Finally, returns diminish as depth increases indicating depth is not the only answer (Table 1).

### 8 CONCLUSIONS

In this work we present Noisy Nodes, a novel regularisation technique for GNNs with particular focus on 3D molecular property prediction. Noisy nodes helps address common challenges around oversmoothed node representations, shows benefits for GNNs of all depths, but in particular improves performance for deeper GNNs. We demonstrate results on challenging 3D molecular property prediction tasks, and some generic GNN benchmark datasets. We believe these results demonstrate Noisy Nodes could be a useful building block for GNNs for molecular property prediction and beyond.

## 9 REPRODUCIBILITY STATEMENT

Code for reproducing OGB-PCQM4M results using Noisy Nodes is available on github, and was prepared as part of a leaderboard submission. `https://github.com/deepmind/deepmind-research/tree/master/ogb_lsc/pcq`.

We provide detailed hyper parameter settings for all our experiments in the appendix, in addition to formulae for computing the encoder and decoder stages of the GNS.

## 10 ETHICS STATEMENT

**Who may benefit from this work?** Molecular property prediction with GNNs is a fast-growing area with applications across domains such as drug design, catalyst discovery, synthetic biology, and chemical engineering. Noisy Nodes could aid models applied to these domains. We also demonstrate on OC20 that our direct state prediction approach is nearly as accurate as learned relaxed approaches at a small fraction of the computational cost, which may support material design which requires many predictions.

Finally, Noisy Nodes could be adapted and applied to many areas in which GNNs are used—for example, knowledge base completion, physical simulation or traffic prediction.

**Potential negative impact and reflection.** Noisy Nodes sees improved performance from depth, but the training of very deep GNNs could contribute to global warming. Care should be taken when utilising depth, and we note that Noisy Nodes settings can be calibrated at shallow depth.

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

## A APPENDIX

The following sections include details on training setup, hyper-parameters, input processing, as well as additional experimental results.

### A.1 ADDITIONAL METRICS FOR OPEN CATALYST IS2RS TEST SET

Relaxation approaches to IS2RS minimise forces with respect to positions, with the expectation that forces at the minimum are close to zero. One metric of such a model's success is to evaluate the forces at the converged structure using ground truth Density Functional Theory calculations and see how close they are to zero. Two metrics are provided by OC20 (Chanussot* et al., 2020) on the IS2RS test set: Force below Threshold (FbT), which is the percentage of structures that have forces below 0.05 eV/Angstrom, and Average Force below Threshold (AFbT) which is FbT calculated at multiple thresholds.

The OC20 project computes test DFT calculations on the evaluation server and presents a summary result for all IS2RS position predictions. Such calculations take 10-12 hours and they are not available for the validation set. Thus, we are not able to analyse the results in Tables 8 and 9 in any further detail. Before application to catalyst screening further work may be needed for direct approaches to ensure forces do not explode from atoms being too close together.

Table 8: OC20 IS2RS Test, Average Force below Threshold %, ↑

| Model | Method | OOD Both | OOD Adsorbate | OOD Catalyst | ID |
|-------|--------|----------|---------------|--------------|-----|
| Noisy Nodes | Direct | 0.09% | 0.00% | 0.29% | 0.54% |

Table 9: OC20 IS2RS Test, Force below Threshold %, ↑

| Model | Method | OOD Both | OOD Adsorbate | OOD Catalyst | ID |
|-------|--------|----------|---------------|--------------|-----|
| Noisy Nodes | Direct | 0.0% | 0.0% | 0.0% | 0.0% |

## A.2 MORE DETAILS ON GNS ADAPTATIONS FOR MOLECULAR PROPERTY PREDICTION.

**Encoder.**

The node features are a learned embedding lookup of the atom type, and in the case of OC20 two additional binary features representing whether the atom is part of the adsorbate or catalyst and whether the atom remains fixed during the quantum chemistry simulation.

The edge features, $e_k$ are the distances $|d|$ featurised using $c$ Radial Bessel basis functions, $\tilde{e}_{RBF,c} = \sqrt{\frac{2}{R}} \frac{\sin(\frac{c\pi}{R}d)}{d}$, and the edge vector displacements, $d$, normalised by the edge distance:

$$e_k = \text{Concat}(\tilde{e}_{RBF,1}(|d|), ..., \tilde{e}_{RBF,c}(|d|), \frac{d}{|d|})$$

Our conversion to fractional coordinates only applied to the vector quantities, i.e. $\frac{d}{|d|}$.

**Decoder**

The decoder consists of two parts, a *graph-level decoder* which predicts a single output for the input graph, and a *node-level decoder* which predicts individual outputs for each node. The graph-level decoder implements the following equation:

$$y = W^{\text{Proc}} \sum_{i=1}^{|V|} \text{MLP}_{\text{Proc}}(a_i^{\text{Proc}}) + b^{\text{Proc}} + W^{\text{Enc}} \sum_{i=1}^{|V|} \text{MLP}_{\text{Enc}}(a_i^{\text{Enc}}) + b^{\text{Enc}}$$

Where $a_i^{\text{Proc}}$ are node latents from the Processor, $a_i^{\text{Enc}}$ are node latents from the Encoder, $W^{\text{Enc}}$ and $W^{\text{Proc}}$ are linear layers, $b^{\text{Enc}}$ and $b^{\text{Proc}}$ are biases, and $|V|$ is the number of nodes. The node-level decoder is simply an MLP applied to each $a_i^{\text{Proc}}$ which predicts $a_i^{\Delta}$.

## A.3 MORE DETAILS ON MPNN FOR OGBG-PCQM4M AND OGBG-MOLPCBA

Our MPNN follows the blueprint of Gilmer et al. (2017). We use $\vec{h}_v^{(t)}$ to denote the latent vector of node $v$ at message passing step $t$, and $\vec{m}_{uv}^{(t)}$ to be the computed message vector for the edge between nodes $u$ and $v$ at message passing step $t$. We define the update functions as:

$$\vec{m}_{uv}^{(t+1)} = \psi_{t+1}\left(\vec{h}_u^{(t)}, \vec{h}_v^{(t)}, \vec{m}_{uv}^{(t)} + \vec{m}_{uv}^{(t-1)}\right) \tag{1}$$

$$\vec{h}_u^{(t+1)} = \phi_{t+1}\left(\vec{h}_u^{(t)}, \sum_{u \in \mathcal{N}_v} \vec{m}_{vu}^{(t+1)}, \sum_{v \in \mathcal{N}_u} \vec{m}_{uv}^{(t+1)}\right) + \vec{h}_u^t \tag{2}$$

Where the message function $\psi_{t+1}$ and the update function $\phi_{t+1}$ are MLPs. We use a "Virtual Node" which is connected to all other nodes to enable long range communication. Out readout function is an MLP. No spatial features are used.

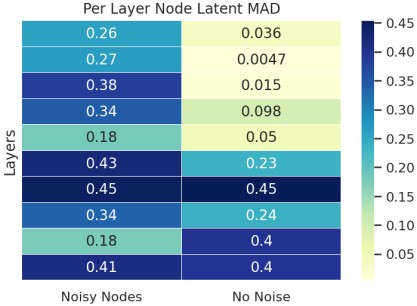

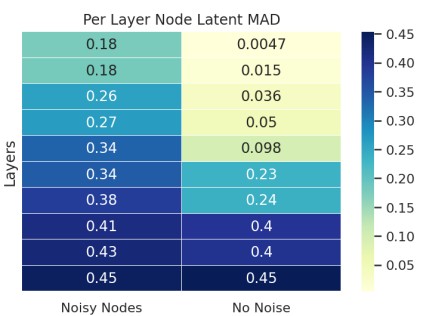

Figure 6: GNS Unsorted MAD per Layer Averaged Over 3 Random Seeds. Evidence of oversmoothing is clear. Model trained on QM9.

Figure 7: GNS Sorted MAD per Layer Averaged Over 3 Random Seeds. The trend is clearer when the MAD values have been sorted. Model trained on QM9.

## A.4 EXPERIMENT SETUP FOR 3D MOLECULAR MODELING

**Open Catalyst.** All training experiments were ran on a cluster of TPU devices. For the Open Catalyst experiments, each individual run (i.e. a single random seed) utilised 8 TPU devices on 2 hosts (4 per host) for training, and 4 V100 GPU devices for evaluation (1 per dataset).

Each Open Catalyst experiment was ran until convergence for up to 200 hours. Our best result, the large 100 layer model requires 7 days of training using the above setting. Each configuration was run at least 3 times in this hardware configuration, including all ablation settings.

We further note that making effective use of our regulariser requires sweeping noise values. These sweeps are dataset dependent and can be carried out using few message passing steps.

**QM9.** Experiments were also run on TPU devices. Each seed was run using 8 TPU devices on a single host for training, and 2 V100 GPU devices for evaluation. QM9 targets were trained between 12-24 hours per experiment.

Following Klicpera et al. (2020b) we define std. MAE as :

$$\text{std. MAE} = \frac{1}{M} \sum_{m=1}^{M} \left( \frac{1}{N} \sum_{i=1}^{N} \frac{|f_\theta^{(m)}(\boldsymbol{X}_i, \boldsymbol{z}_i) - \hat{t}_i^{(m)}|}{\sigma_m} \right)$$

and logMAE as:

$$\text{logMAE} = \frac{1}{M} \sum_{m=1}^{M} \log \left( \frac{1}{N} \sum_{i=1}^{N} \frac{|f_\theta^{(m)}(\boldsymbol{X}_i, \boldsymbol{z}_i) - \hat{t}_i^{(m)}|}{\sigma_m} \right)$$

with target index $m$, number of targets $M = 12$, dataset size $N$, ground truth values $\hat{t}^{(m)}$, model $f_\theta^{(m)}$, inputs $\boldsymbol{X}_i$ and $\boldsymbol{z}_i$, and standard deviation $\sigma_m$ of $\hat{t}^{(m)}$.

## A.5 OVER SMOOTHING ANALYSIS FOR GNS

In addition to Figure 2, we repeat the analysis with a mean MAD over 3 seeds 7. Furthermore we remove the sorting layer by MAD value and find the trend holds.

## A.6 NOISE ABLATIONS FOR OGBG-MOLPCBA

We conduct a noise ablation on the random flipping noise for OGBG-MOLPCBA with an 8 layer MPNN + Virtual Node, and find that our model is not very sensitive to the noise value (Table 10), but degrades from 0.1.

| Flip Probability | Mean AP |
|---|---|
| 0.01 | 27.8% +- 0.002 |
| 0.03 | 27.9% +- 0.003 |
| 0.05 | **28.1**% +- 0.001 |
| 0.1 | 28.0% +- 0.003 |
| 0.2 | 27.7% +- 0.002 |

Table 10: OGBG-MOLPCBA Noise Ablation

| | Mean AP |
|---|---|
| MPNN Without DropEdge | 27.4% $\pm$ 0.002 |
| MPNN With DropEdge | 27.5% $\pm$ 0.001 |
| MPNN + DropEdge + Noisy Nodes | **27.8**% $\pm$ 0.002 |

Table 11: OGBG-MOLPCBA DropEdge Ablation

## A.7 DropEdge & DropNode Ablations for OGBG-MOLPCBA

We conduct an ablation with our 16 layer MPNN using DropEdge at a rate of 0.1 as an alternative approach to improving oversmoothing and find it does not improve performance for ogbg-molpcba (Table 11), similarly we find DropNode (Table 12) does not improve performance. In addition, we find that these two methods can't be combined well together, reaching a performance of 27.0% $\pm$ 0.003. However, both methods can be combined advantageously with Noisy Nodes.

We also measure the MAD of the node latents for each layer and find the indeed Noisy Nodes is more effective at addressing oversmoothing in Figure 8.

## A.8 Training Curves for OC20 Noisy Nodes Ablations Demonstrating Overfitting

Figure 9

| | Mean AP |
|---|---|
| MPNN With DropNode | 27.5% $\pm$ 0.001 |
| MPNN Without DropNode | 27.5% $\pm$ 0.004 |
| MPNN + DropNode + Noisy Nodes | **28.2**% $\pm$ 0.005 |

Table 12: OGBG-MOLPCBA DropNode Ablation

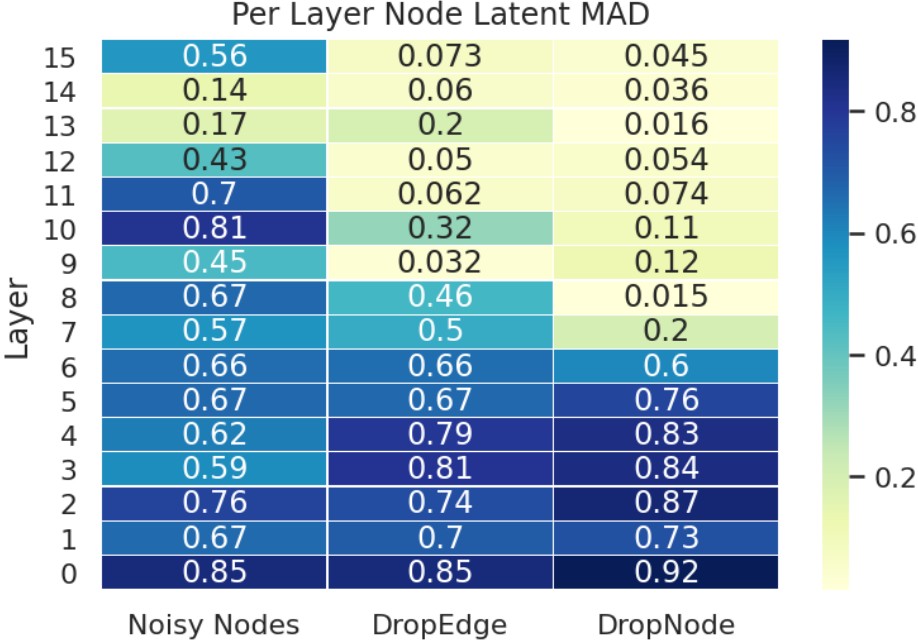

Figure 8: Comparison of the effect of techniques to address oversmoothing on MPNNs. Whilst Some effect can be seen from DropEdge and DropNode, Noisy Nodes is significantly better at preserving per node diversity.

### A.9 PSEUDOCODE FOR 3D MOLECULAR PREDICTION TRAINING STEP

---

**Algorithm 1:** Noisy Nodes Training Step

---

$G = (V, E, g)$ // Input graph
$\tilde{G} = G$ // Initialize noisy graph
$\lambda$ // Noisy Nodes Weight
**if** *not_provided(V')* **then**
  |   $V' \leftarrow V$
**end**
**if** *predict_differences* **then**
  |   $\Delta = \{v'_i - v_i | i \in 1, \ldots, |V|\}$
**end**
**for each** $i \in 1, \ldots, |V|$ **do**
  |   $\sigma_i = $ sample_node_noise(shape_of($v_i$));
  |   $\tilde{v}_i = v_i + \sigma_i$;
  |   **if** *predict_differences* **then**
  |    |   $\tilde{\Delta}_i = \Delta_i - \sigma_i$;
  |   **end**
**endfor**
$\tilde{E} = $ recompute_edges($\tilde{V}$);
$\hat{G}' = $ GNN($\tilde{G}$);
**if** *predict_differences* **then**
  |   $V' = \tilde{\Delta}_i$;
**end**
Loss $= \lambda$ NoisyNodesLoss($\hat{G}', V'$) + PrimaryLoss($\hat{G}', V'$));
Loss.minimise()

---

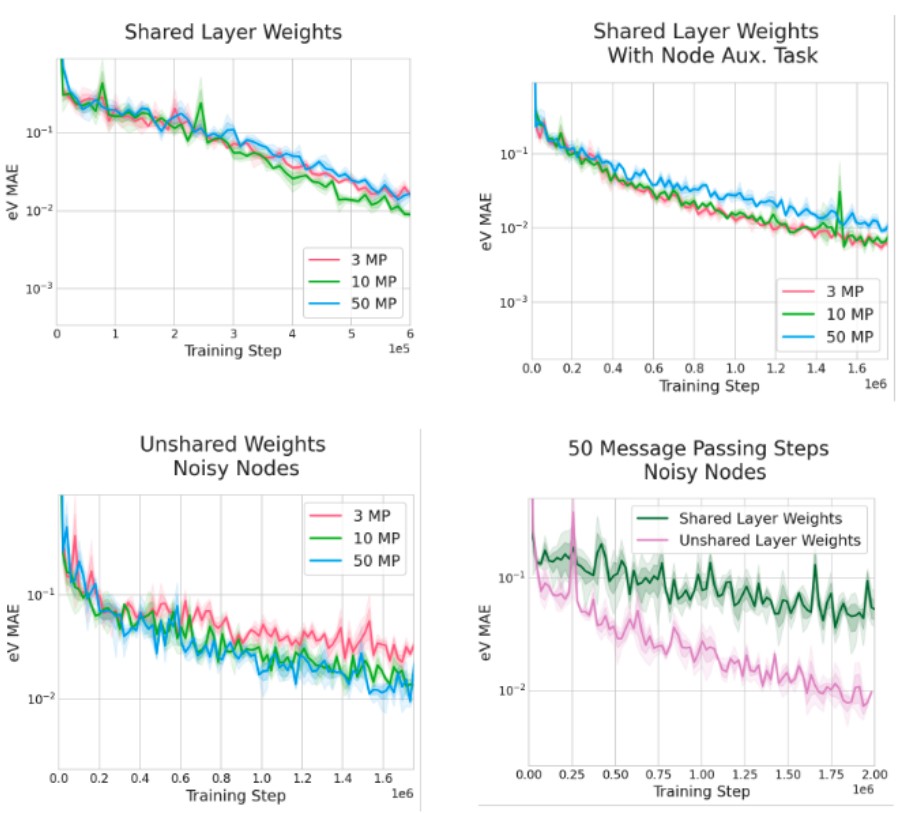

Figure 9: Training curves to accompany Figure 3. This demonstrates that even as the validation performance is getting worse, training loss is going down, indicating overfitting.

Table 13: Open Catalyst training parameters.

| Parameter | Value or description |
|---|---|
| Optimiser | Adam with warm up and cosine cycling |
| $\beta_1$ | 0.9 |
| $\beta_2$ | 0.95 |
| Warm up steps | $5e5$ |
| Warm up start learning rate | $1e-5$ |
| Warm up/cosine max learning rate | $1e-4$ |
| Cosine cycle length | $5e6$ |
| Loss type | Mean squared error |
| Batch size | Dynamic to max edge/node/graph count |
| Max nodes in batch | 1024 |
| Max edges in batch | 12800 |
| Max graphs in batch | 10 |
| MLP number of layers | 3 |
| MLP hidden sizes | 512 |
| Number Bessel Functions | 512 |
| Activation | shifted softplus |
| message passing layers | 50 |
| Group size | 10 |
| Node/Edge latent vector sizes | 512 |
| Position noise | Gaussian ($\mu = 0, \sigma = 0.3$) |
| Parameter update | Exponentially moving average (EMA) smoothing |
| EMA decay | 0.9999 |
| Position Loss Co-efficient | 1.0 |

### A.10 TRAINING DETAILS

Our code base is implemented in JAX using Haiku and Jraph for GNNs, and Optax for training (Bradbury et al., 2018; Babuschkin et al., 2020; Godwin* et al., 2020; Hennigan et al., 2020). Model selection used early stopping.

All results reported as an average of 10 random seeds. OGBG-PCQM4M & OGBG-MOLPCBA were trained with 16 TPUs and evaluated with a single V100 GPU. OGBN-Arxiv was trained and evalated with a single TPU

**3D Molecular Prediction**

We minimise the mean squared error loss on mean and standard deviation normalised targets and use the Adam (Kingma & Ba, 2015) optimiser with warmup and cosine decay. For OC20 IS2RE energy prediction we subtract a learned reference energy, computed using an MLP with atom types as input.

For the GNS model the node and edge latents as well as MLP hidden layers were sized 512, with 3 layers per MLP and using shifted softplus activations throughout. OC20 & QM9 Models were trained on 8 TPU devices and evaluated on a single V100 GPUs. We provide the full set of hyper-parameters and computational resources used separately for each dataset in the Appendix. All noise levels were determined by sweeping a small range of values ($\approx 10$) informed by the noised feature covariance.

**Non Spatial Tasks**

### A.11 HYPER-PARAMETERS

**Open Catalyst.** We list the hyper-parameters used to train the default Open Catalyst experiment. If not specified otherwise (e.g. in ablations of these parameters), experiments were ran with this configuration.

Table 14: QM9 training parameters.

| Parameter | Value or description |
|---|---|
| Optimiser | Adam with warm up and cosine cycling |
| $\beta_1$ | 0.9 |
| $\beta_2$ | 0.95 |
| Warm up steps | $1e4$ |
| Warm up start learning rate | $3e-7$ |
| Warm up/cosine max learning rate | $1e-4$ |
| Cosine cycle length | $2e6$ |
| Loss type | Mean squared error |
| Batch size | Dynamic to max edge/node/graph count |
| Max nodes in batch | 256 |
| Max edges in batch | 4096 |
| Max graphs in batch | 8 |
| MLP number of layers | 3 |
| MLP hidden sizes | 1024 |
| Number Bessel Funtions | 512 |
| Activation | shifted softplus |
| message passing layers | 10 |
| Group Size | 10 |
| Node/Edge latent vector sizes | 512 |
| Position noise | Gaussian ($\mu = 0, \sigma = 0.02$) |
| Parameter update | Exponentially moving average (EMA) smoothing |
| EMA decay | 0.9999 |
| Position Loss Coefficient | 0.1 |

Dynamic batch sizes refers to constructing batches by specifying maximum node, edge and graph counts (as opposed to only graph counts) to better balance computational load. Batches are constructed until one of the limits is reached.

Parameter updates were smoothed using an EMA for the current training step with the current decay value computed through $decay = min(decay, (1.0 + step)/(10.0 + step))$. As discussed in the evaluation, best results on Open Catalyst were obtained by utilising a 100 layer network with group size 10.

**QM9** Table 14 lists QM9 hyper-parameters which primarily reflect the smaller dataset and geometries with fewer long range interactions. For $U_0$, $U$, $H$ and $G$ we use a slightly larger number of graphs per batch - 16 - and a smaller position loss co-efficient of 0.01.

**OGBG-PCQM4M** Table 15 provides the hyper parameters for OGBG-PCQM4M.

**OGBG-MOLPCBA** Table 16 provides the hyper parameters for the OGBG-MOLPCBA experiments.

**OGBN-ARXIV** Table 17 provides the hyper parameters for the OGBN-Arxiv experiments.

Table 15: OGBG-PCQM4M Training Parameters.

| Parameter | Value or description |
|---|---|
| Optimiser | Adam with warm up and cosine cycling |
| $\beta_1$ | 0.9 |
| $\beta_2$ | 0.95 |
| Warm up steps | $5e4$ |
| Warm up start learning rate | $1e-5$ |
| Warm up/cosine max learning rate | $1e-4$ |
| Cosine cycle length | $5e5$ |
| Loss type | Mean absolute error |
| Reconstruction type | Softmax Cross Entropy |
| Batch size | Dynamic to max edge/node/graph count |
| Max nodes in batch | 20,480 |
| Max edges in batch | 8,192 |
| Max graphs in batch | 512 |
| MLP number of layers | 2 |
| MLP hidden sizes | 512 |
| Activation | relu |
| Node/Edge latent vector sizes | 512 |
| Noisy Nodes Category Flip Fate | 0.05 |
| Parameter update | Exponentially moving average (EMA) smoothing |
| EMA decay | 0.999 |
| Reconstruction Loss Coefficient | 0.1 |

Table 16: OGBG-MOLPCBA Training Parameters.

| Parameter | Value or description |
|---|---|
| Optimiser | Adam with warm up and cosine cycling |
| $\beta_1$ | 0.9 |
| $\beta_2$ | 0.95 |
| Warm up steps | $1e4$ |
| Warm up start learning rate | $1e-5$ |
| Warm up/cosine max learning rate | $1e-4$ |
| Cosine cycle length | $1e5$ |
| Loss type | Softmax Cross Entropy |
| Reconstruction loss type | Softmax Cross Entropy |
| Batch size | Dynamic to max edge/node/graph count |
| Max nodes in batch | 20,480 |
| Max edges in batch | 8,192 |
| Max graphs in batch | 512 |
| MLP number of layers | 2 |
| MLP hidden sizes | 512 |
| Activation | relu |
| Batch Normalization | Yes, after every hidden layer |
| Node/Edge latent vector sizes | 512 |
| Dropnode Rate | 0.1 |
| Dropout Rate | 0.1 |
| Noisy Nodes Category Flip Fate | 0.05 |
| Parameter update | Exponentially moving average (EMA) smoothing |
| EMA decay | 0.999 |
| Reconstruction Loss Coefficient | 0.1 |

Table 17: OGBG-ARXIV Training Parameters.

| Parameter | Value or description |
|---|---|
| Optimiser | Adam with warm up and cosine cycling |
| $\beta_1$ | 0.9 |
| $\beta_2$ | 0.95 |
| Warm up steps | 50 |
| Warm up start learning rate | $1e-5$ |
| Warm up/cosine max learning rate | $1e-3$ |
| Cosine cycle length | $12,000$ |
| Loss type | Softmax Cross Entropy |
| Reconstruction loss type | Mean Squared Error |
| Batch size | Full graph |
| MLP number of layers | 1 |
| Activation | relu |
| Batch Normalization | Yes, after every hidden layer |
| Node/Edge latent vector sizes | 256 |
| Dropout Rate | 0.5 |
| Noisy Nodes Input Dropout | 0.05 |
| Reconstruction Loss Coefficient | 0.1 |

