# OpenReview forum: "Simple GNN Regularisation for 3D Molecular Property Prediction and Beyond"
_ICLR.cc/2022/Conference — ICLR 2022 Poster_

### Official Review · Reviewer_7LGq · 2021-10-27

**Correctness:** 3
**Technical Novelty And Significance:** 2
**Empirical Novelty And Significance:** 2
**Recommendation:** 6
**Confidence:** 4

**Main Review:**


Strength:
- The method is pretty simple, but the results are very positive on several benchmarks, which has achieved improvements over several baselines.

Weakness:

The writing needs to be improved, especially for the method section and the experiment section. I find many important details are missing. For example. How is the noise-enhanced auxiliary loss incorporated into the 3D prediction task? How are the two aspects balanced?

The claimed relative improvements need to be broken down in Table 1-4. For
example, in table 1, there are many different factors that may contribute to
the claimed improvements over existing baselines: is the improvement due to
the advantage of GNS over baselines, or the changes made to GNS, or the core
idea of noise injection?

Some important baselines are missing. Two categories of baselines need to be
added: (1) There are several existing works on injecting adversarial losses
into graph neural networks, which can be also combined with GNS for 3d
molecular prediction. How does this proposed method compare with those
existing methods? (2) For 3D molecular modeling, there are other more powerful
equivariant models, such as Tensor Field Network, Se(3)-Transformers, GemNet.
Without including those baselines, it is hard to conclude how this method
advances the SOTA for 3d molecular property prediction.

It's better to perform sensitivity analysis to the injected noise. How are the
noises chosen for different datasets, and how sensitive is the model to such
noise?



**Summary Of The Paper:**

This paper proposes a noise-injected training mechanism for graph neural network. The idea is to perturb the node attributes and add an auxiliary loss for the noise-correction task. The authors. Implementing this strategy for the GNS model And observed improvements on several 3D molecular prediction tasks.


**Summary Of The Review:**

The proposed idea is intuitive and sound, and the results are positive. However,  important baselines and detailed experiments needs to be added to support the claims and justify the contributions. The writing also needs to be improved.

---

> ### Author Response · Authors · 2021-11-15
> **Response with Additional Analysis & Experiments (1/2)**
>
> Dear Reviewer,
>
> Thank you for your comments. You have raised some valid points, and we have conducted additional experiments and edits to the manuscript to address your concerns. We hope our changes have improved your view of the manuscript and you would consider raising your score.
>
> **The writing needs to be improved, especially for the method section and the experiment section. I find many important details are missing. For example. How is the noise-enhanced auxiliary loss incorporated into the 3D prediction task? How are the two aspects balanced?**
>
> Thank you for this comment, we agree this phrasing could be improved.
>
> The auxiliary loss (noisy_nodes)  is weighted by a constant coefficient \lambda, and summed with the primary loss. The value of this weighting coefficient is set by hyperparameter, and the values are given in the appendix.
>
> $Loss = \lambda \text{NoisyNodesLoss}(\hat{G}', V') + \text{PrimaryLoss}(\hat{G}', V'))$
>
> In addition, we have added Algorithm 1 in the Appendix which we hope clarifies these points further.
>
>
> **The claimed relative improvements need to be broken down in Table 1-4. For example, in table 1, there are many different factors that may contribute to the claimed improvements over existing baselines: is the improvement due to the advantage of GNS over baselines, or the changes made to GNS, or the core idea of noise injection?**
>
> Thank you for this comment. We agree these tables can be improved. For ease of reference we present the binary comparisons below.
>
> In Table 1, row 1 and row 3 compare the GNS with GNS + Noisy Nodes, and a significant improvement is seen. We repeat these figures below for ease of comparison:
>
> |                   | In Distribution Validation IS2RE Performance (eV MAE) |
> |-------------------|-------------------------------------------------------|
> | GNS               | 0.54 +- 0.00                                          |
> | GNS + Noisy Nodes | 0.48 +- 0.00                                          |
>
>
>
> In Table 3 row 1 is GNS, and line 3 GNS + Noisy Nodes and again a significant improvement is seen. We have repeated the results below for ease of comparison:
>
> |                   | In Distribution Validation IS2RS Performance (ADwT, Higher Better) |
> |-------------------|--------------------------------------------------------------------|
> | GNS               | 0.40.0 +- 0.00                                                     |
> | GNS + Noisy Nodes | 0.46.9 +- 0.00                                                     |
>
> This, however, is not the only dataset we evaluate GNS on. For QM9 we show that adding noisy nodes leads to a relative improvement of 23.1% in the std. MAE metric (Table 5, row 1 and 3). Without this improvement, the GNS would be the worst but one out of the benchmark results, demonstrating the importance of using Noisy Nodes.
>
> Finally, we establish in the remaining sections performance improvements for 2 more architectures on 3 different datasets. We hope this analysis helps persuade you that the gains are due to Noisy Nodes.
>
>
> **Some important baselines are missing. Two categories of baselines need to be added: (1) There are several existing works on injecting adversarial losses into graph neural networks, which can be also combined with GNS for 3d molecular prediction. How does this proposed method compare with those existing methods?**
>
> Thank you for this comment, we agree we could more clearly reference other noise injection work
>
> We have referenced FLAG, a method for injecting adversarial noise in our related work. We are not aware of further work on adversarial noise injection to improve results, but would be happy to include them.
>
> The key difference we see in these methods is that FLAG is a method for generating noise, whilst Noisy Nodes is a way to use this noise by adding an additional loss term, which we show is very important for performance improvement (Table 3). We have focused on simple noise generation to demonstrate the effectiveness of our method, but we believe FLAG could be a very effective alternative that combines well with Noisy Noise.
>
> We have updated our text to make this comparison clearer.

---

> > ### Author Response · Authors · 2021-11-15
> > **Response with Additional Analysis & Experiments (2/2)**
> >
> > **(2) For 3D molecular modeling, there are other more powerful equivariant models, such as Tensor Field Network, Se(3)-Transformers, GemNet. Without including those baselines, it is hard to conclude how this method advances the SOTA for 3d molecular property prediction.**
> >
> > Thank you for this comment. We agree the contribution of our paper could be made clearer.
> >
> > Since Noisy Nodes can be combined with the models you cite, we expect that the combination of Noisy Nodes with these more powerful models would attain even better results. We outline how to approach this in section 6.4.
> >
> > Our contribution is to show how baseline models that do not perform well out of the box (i.e. GNS) can be used with Noisy Nodes to achieve competitive results. We do not claim that Noisy Nodes (as regularisation technique) + GNS advances SOTA in general, but we were very happy to see improved results on OC20 against existing reported models. We are not aware of other models that have performed as well on this task. We have updated the text of our paper to reflect these points more clearly.
> >
> > Following your comment tried again to find better performing QM9 models and our understanding is that PaiNN (a paper we have benchmarked in our manuscript) is SOTA. For OC20, Gemnet has been trained on the relaxation task (we train on the "direct" tasl) with impressive results, and we have updated our manuscript to point this out. However, we believe we have reported the most competitive results for both of the 3D molecular property prediction datasets we have evaluated on.
> >
> > Finally, thank you for referencing the SE(3) transformer. We did not include it because it does not report all results on QM9, but we compare the subset to  GNS + Noisy Nodes below with the MAE measure and note than GNS + Noisy Nodes performs better.
> >
> > |                   | alpha | gap  | homo | lumo | mu    | cv    |
> > |-------------------|-------|------|------|------|-------|-------|
> > | SE(3) Transformer | 0.142 | 53   | 40   | 38   | 0.064 | .101  |
> > | GNS + Noisy Nodes | 0.052 | 28.6 | 20.4 | 18.6 | 0.025 | 0.025 |
> >
> > **It's better to perform sensitivity analysis to the injected noise. How are the noises chosen for different datasets, and how sensitive is the model to such noise?**
> >
> > Thank you for this suggestion, we have conducted an ablation on noise for OGBG-MOLPCBA for an 16 layer MPNN + Virtual Node and provide it in the appendix in our paper. We repeat the results below, which show that our model is not very sensitive to the noise value.
> >
> >
> > | Random Flip Probability | Mean AP           |
> > |-------------------------|-------------------|
> > | 0.01                   | 27.8% +- 0.002|
> > | 0.03                    | 27.9% +- 0.003|
> > | 0.05                    |28.1%+- 0.001 |
> > | 0.1                     | 28.0% +- 0.003 |
> > | 0.2                     | 27.7% +- 0.002 |
> >
> > In addition we show that Noisy Nodes can compose with alternatives ways to address oversmoothing. We have now included an ablation with DropNode & DropEdge applied to OGB-MOLPCBA  with a 16 layer MPNN + Virtual Node in the Appendix. We repeat the results here for ease of comparison.
> >
> >
> > |                      | Mean AP              |
> > |----------------------|----------------------|
> > | MPNN | 27.5%±0.001 |
> > | MPNN + Drop Node   | 27.5%±0.004|
> > | MPNN + Drop Node  + Noisy Nodes | 28.2%±0.005|
> >
> >
> > Above we see that DropNode does not improve the performance of our network, However, Drop Node combines favorably with Noisy Nodes.
> >
> > We find the same effect for DropEdge:
> >
> > |                      | Mean AP              |
> > |----------------------|----------------------|
> > | MPNN | 27.4%±0.002 |
> > | MPNN + Drop Edge  | 27.5%±0.001 |
> > | MPNN + Drop Edge + Noisy Nodes | 27.8%±0.002 |
> >
> > Interestingly, we find that DropEdge and DropNode do not combine well together, reaching a performance of 27.0%±0.003. However, Noisy Nodes combined with both methods reaches a Mean AP of 27.6% 0 0.0004
> >
> > As part of this study we have also made a small change to the main text results for OGBG-MOLPCBA.  We have now broken DropNode & DropEdge out into an ablation study in the appendix.
> >
> > Finally, in Figure 8 of the updated text we analyse oversmoothing in the node latents for each of the proposed methods. In this analysis we find that Noisy Nodes is significantly more successful at preserving per node latent diversity than either DropEdge or DropNode for MPNNs.

---

> ### Author Response · Authors · 2021-11-24
> **Please let us know if you have any further questions or comments.**
>
> We believe we have addressed your concerns, but please let us know if you have other questions or comments and we will be very happy to respond.

---

### Official Review · Reviewer_Y3os · 2021-11-02

**Correctness:** 3
**Technical Novelty And Significance:** 2
**Empirical Novelty And Significance:** 3
**Recommendation:** 6
**Confidence:** 4

**Main Review:**

**Strengths**
- The simplicity of the proposal.
- Solid empirical results.

**Weaknesses**
- It is not entirely clear where the gain is coming from (W1);
- No comparison against other ways to achieve deep GNNs, i.e., ways to tackle over-smoothing (W2);
- Limited technical novelty with non-concrete claims (W3);
- Experimental setup: not clear why the authors focus on molecules (W4).

**Detailed comments**
- (W1) The authors fail to show that GNS (same setup but without noisy nodes and additional loss) actually suffers from over-smoothing and overfitting (the paper's initial motivation). Figure 3 is not enough to demonstrate overfitting as it only exhibits validation curves and no training ones. Also, Figure 3A shows that the performance of 3MP and 50MP are very close --- can we say over-smoothing is an issue here? Additionally, it seems the authors already apply other regularization strategies (dropout, early stopping, etc.) --- aren't they enough to avoid overfitting?

- (W2) Since "Noisy Nodes" is introduced as a new simple regularization technique, I believe it is crucial to assess how the proposal compares with alternatives to handle over-smoothing in GNNs, such as PairNorm [1], long-range residual connections [2], and DropEdge [3].

- (W3) The authors provide little support to their claims in Section 4, which are supposed to explain why the method works. For instance, the authors say that "the GNN learns to go from low probability graphs to high probability graphs". However, it is hard to guarantee that the noisy graphs lie in low-density regions, especially if we consider high-dimensional spaces.

- (W4) Is there any aspect of the method that makes it particularly tailored to molecular data or regression tasks? The gains on Arxiv seem to be less significant compared to those in other datasets.

**Additional observations**
* Noise has also been used to increase the expressive power of GNNs [4]. It should probably be mentioned in the paper.
* Is there any particular reason for picking base GNNs with virtual nodes in Section 6?
* It is weird to see tables full of numerical results with 0 std (Tables 1 and 3).
* Typos: "..adversarial noise during to input.." (page 2); " range of of " (page 4); "which report which report" (page 9).
* Reference Zonghan Wu et al. is duplicated.

[1] Lingxiao Zhao, Leman Akoglu. PairNorm: Tackling Oversmoothing in GNNs, ICLR, 2020.

[2] Ming Chen et al., Simple and Deep Graph Convolutional Networks. ICML, 2020.

[3] Yu Rong et al. Dropedge: Towards deep graph convolutional networks on node classification. ICLR, 2020.

[4] Ryoma Sato, Makoto Yamada, and Hisashi Kashima. Random features strengthen graph neural networks. arXiv preprint, 2020.



**Summary Of The Paper:**

The paper proposes a new regularization method for tackling both over-smoothing and overfitting in GNNs. The main idea is to add node-level noise to input graphs and augment the loss function with a denoising term. The authors run experiments on three benchmarks for 3D molecular property prediction and three datasets (with no 3D features) from Open Graph Benchmark. Overall,  GNS (Graph Network Simulator) and vanilla GCNs are used as base GNNs. The proposed method outperforms the existing approaches.


**Summary Of The Review:**

Overall, I like the simplicity of the proposal and the fact the paper tackles relevant issues in GNNs. Also, the paper shows good empirical results. However, my main concerns revolve around limited technical novelty, lack of experiments to validate some intuitions behind the method, and comparison against related methods.

---

> ### Author Response · Authors · 2021-11-15
> **Response with Analysis & Experiments (1/2)**
>
> Dear Reviewer,
>
> Thank you for your helpful review. You’ve raised some important points that we have addressed with additional experiments and analysis, and have updated the paper accordingly. We hope that if our additional analysis and experiments have sufficiently addressed your concerns you will consider raising your score.
>
> **“The authors fail to show that GNS (same setup but without noisy nodes and additional loss) actually suffers from over-smoothing and overfitting (the paper's initial motivation)”**
>
> Thank you for raising this, we agree that our paper could explain this better. In Figure 2 we present some analysis of the diversity of GNS node latents using Mean Absolute Distance, of the GNS trained on QM9 with and without Noisy Nodes. This shows that node diversity decreases sharply for the GNS without Noisy Nodes. We agree that this analysis could be more clearly sign-posted as the GNS model and we have updated our text to make this analysis clearer.
>
> In addition, we have conducted further analysis to demonstrate over-smoothing for GNS. In Figures 6 & 7 in the appendix we demonstrate the MAD metric over a mean of three different seeds, with the same results. We hope this analysis helps persuade you that GNS suffers from oversmoothing.
>
> **“Figure 3 is not enough to demonstrate overfitting as it only exhibits validation curves and no training ones.”**
>
> Thank you for this comment. We should have explained that we observed training loss continuing to decrease. We have now provided training curves in the Appendix Figure 9 that show that training loss continues to decrease as validation error increases.
>
> **“ Also, Figure 3A shows that the performance of 3MP and 50MP are very close --- can we say over-smoothing is an issue here? “**
>
> Thank you for this comment, we agree this analysis could be improved.  A symptom of over-smoothing is that there is no improvement in performance as depth increases. We believe the fact that 3 MP and 50 MP are close together supports the claim that oversmoothing is taking place. We have updated our text to make this point clearer.
>
> **“Additionally, it seems the authors already apply other regularization strategies (dropout, early stopping, etc.) --- aren't they enough to avoid overfitting?”**
>
> Thank you for this comment, we agree this analysis could be improved. Whilst we do use other regularization strategies, we found that combining them with Noisy Nodes improved results. For example, in Figure 3 Noisy Nodes improves the minimum validation result, which would combine well with early stopping. Dropout was found to be helpful on some datasets, but always composed well with Noisy Nodes. All values were chosen with a small (<5) hyper parameter search, and the chosen hyperparameters are all listed in the appendix. We have updated our text with these points.
>
> **“Since "Noisy Nodes" is introduced as a new simple regularization technique, I believe it is crucial to assess how the proposal compares with alternatives to handle over-smoothing in GNNs, such as PairNorm [1], long-range residual connections [2], and DropEdge [3].”**
>
> Thank you for this criticism. While we have cited alternative methods in the related works and noted some differences, we agree that a more thorough analysis of different methods would be useful.
> PairNorm, DropEdge and long-range residual connections are orthogonal and complementary to our approach - you can combine them to improve your network, or see what works best for your task. To demonstrate this, we use the example of DropEdge & DropNode.
>
> We have now included an ablation with DropNode & DropEdge applied to OGB-MOLPCBA  with a 16 layer MPNN + Virtual Node in the Appendix. We repeat the results here for ease of comparison.
>
>
> |                      | Mean AP              |
> |----------------------|----------------------|
> | MPNN | 27.5%±0.001 |
> | MPNN + Drop Node   | 27.5%±0.004|
> | MPNN + Drop Node  + Noisy Nodes | 28.2%±0.005|
>
>
> Above we see that DropNode does not improve the performance of our network, However, Drop Node combines favorably with Noisy Nodes.
>
> We find the same effect for DropEdge:
>
> |                      | Mean AP              |
> |----------------------|----------------------|
> | MPNN | 27.5%±0.002 |
> | MPNN + Drop Edge  | 27.5%±0.001 |
> | MPNN + Drop Edge + Noisy Nodes | 27.8%±0.002 |
>
> Interestingly, we find that DropEdge and DropNode do not combine well together, reaching a performance of 27.0%±0.003.However, Noisy Nodes combined with both methods reaches a Mean AP of 27.6% 0 0.0004
>
> Finally, in Figure 8 of the updated text we analyse oversmoothing in the node latents for each of the proposed methods. In this analysis we find that Noisy Nodes is significantly more successful at preserving per node latent diversity than either DropEdge or DropNode for MPNNs.
>
> We note we have mistakenly omitted the reference for PairNorm which we will update for camera ready along with some additional text for comparison.

---

> > ### Author Response · Authors · 2021-11-15
> > **Response with Analysis & Experiments (2/2)**
> >
> > **“The authors provide little support to their claims in Section 4, which are supposed to explain why the method works. For instance, the authors say that "the GNN learns to go from low probability graphs to high probability graphs". However, it is hard to guarantee that the noisy graphs lie in low-density regions, especially if we consider high-dimensional spaces.”**
> >
> > Thank you for this comment, we agree this section could be improved, and we have updated our text.
> >
> > While we agree that it is hard to guarantee that a noisy graph does not lie on the data manifold, we were inspired by established works [1, 2, 3] that use comparable analysis to motivate learning representations and score based models. These references have now been sign-posted in the text, and we have relaxed our claims, indicating that it is speculative that this is the reason the method works.
> >
> > For other claims in section 4 related to oversmoothing please see our response above.
> >
> > [1] Vincent, Pascal, H. Larochelle, Isabelle Lajoie, Yoshua Bengio and Pierre-Antoine Manzagol. “Stacked Denoising Autoencoders: Learning Useful Representations in a Deep Network with a Local Denoising Criterion.” J. Mach. Learn. Res. 11 (2010): 3371-3408.
> >
> > [2]Vincent, Pascal. “A Connection Between Score Matching and Denoising Autoencoders.” Neural Computation 23 (2011): 1661-1674.
> >
> > [3] Song, Yang and Stefano Ermon. “Generative Modeling by Estimating Gradients of the Data Distribution.” ArXiv abs/1907.05600 (2019): n. pag.
> >
> >
> > **“Is there any aspect of the method that makes it particularly tailored to molecular data or regression tasks? The gains on Arxiv seem to be less significant compared to those in other datasets.”**
> >
> > Thank you for this comment. There is nothing in Noisy Nodes tailored to molecular data or regression, but we agree that our motivations for 3D molecular data could be made clearer.
> >
> > We included 3D molecular prediction because of the clear energy interpretation of adding noise (covered in section 4). We believe this is the easiest way to understand the benefits of noisy nodes - building a model mapping high energy (low probability) 3D molecular configurations to low energy  (high probability) configurations. Because our data is at a low energy local minimum, we can say with high confidence that noised molecules are higher energy.
> >
> > Not all of the non-spatial tasks are regression. For example, OGBG-MOLPCBA is a classification task. Noisy Nodes also uses a classification loss for OGBG-MOLPCBA and OGBG-PCQM4M because the input features are categorical.
> >
> > While the improvements are more minor on Arxiv, this dataset has a small gap between the very best performing results (74.31%) (which use many additional methods, such as label smoothing and self knowledge distillation) and the baseline GCN results (71.74%), and so gains are expected to be smaller.
> >
> > We have updated the text in our paper to make these points clearer.
> >
> > **Noise has also been used to increase the expressive power of GNNs [4]. It should probably be mentioned in the paper.**
> >
> > We have added this to our paper.
> >
> > **Is there any particular reason for picking base GNNs with virtual nodes in Section 6?**
> >
> > The majority of the baselines for these datasets include Virtual Nodes, so we included it to aid comparison.
> >
> > **Typos: "..adversarial noise during to input.." (page 2); " range of of " (page 4); "which report which report" (page 9).**
> >
> > Thank you, this has been corrected.
> >
> >
> > **Reference Zonghan Wu et al. is duplicated.**
> >
> > Thank you, this has been corrected.
> >
> >
> > In addition to the specific comments, we have conducted an additional noise ablation that shows that the model is not very sensitive to the noise value chosen. We have conducted an ablation on noise for OGBG-MOLPCBA and provide it in the appendix in our paper. We repeat the results below, which show that our model gains benefits from using noise at multiple different levels, and seems to degrade slightly as we go above 0.05. In general we found trying a small set of values allowed us to find a good value.
> >
> > | Random Flip Probability | Mean AP           |
> > |-------------------------|-------------------|
> > | 0.01                   | 27.8% +- 0.002|
> > | 0.03                    | 27.9% +- 0.003|
> > | 0.05                    |28.1%+- 0.001 |
> > | 0.1                     | 28.0% +- 0.003 |
> > | 0.2                     | 27.7% +- 0.002 |

---

> ### Author Response · Authors · 2021-11-24
> **Please let us know if you have other questions or comments and we will be very happy to respond.**
>
> We believe we have addressed your concerns, but please let us know if you have other questions or comments and we will be very happy to respond.

---

> > ### Comment · Reviewer_Y3os · 2021-11-29
> > **Response to authors**
> >
> > I want to thank the authors for their feedback, which addressed most of my concerns. I am increasing my score to 6.

---

### Official Review · Reviewer_yQtf · 2021-11-02

**Correctness:** 4
**Technical Novelty And Significance:** 3
**Empirical Novelty And Significance:** Not applicable
**Recommendation:** 6
**Confidence:** 4

**Main Review:**

Strengths
1. The NN method shows improved performance on a wide variety of tasks for different GNN architectures. The paper includes experiments with both graph level prediction tasks as well as node level prediction tasks and NN is shown to improve for both kinds of tasks.
2. It has generally been hard to train very deep GNNs for many problems because of oversmoothing problem. The NN method is shown to work well with upto a depth of 50 in the paper.
3. The paper clearly demonstrates that a model trained with NN is able to overcome oversmoothing by showing the MAD statistic at each layer (fig 2).

Weaknesses
1. NN adds additional hyper parameters like noise standard deviation and weight of the auxiliary loss. From reading the paper, it is unclear how sensitive the final performance is to these hyper-params. It would be good to shed some light on this because the practical utility of a method like this depends on the sensitivity to such hyper-params.
2. In table 7, the 16 layer model is only shown with NN. Please add a column to show performance without NN.
3. While the current results are very impressive, some of the best models for molecular prediction use higher order terms (e.g. Senet, Dimenet++, Gemnet). It is unclear if the NN method can work with such models.

A few more things that could be added to the paper to improve it:
1. The paper shows benefits of scaling the depth up to 50 layers. It would be good to show what happens beyond this depth. For example, does performance keep improving and plateau at some point, or does it deteriorate beyond a limit?
2. NN can be seen as both a regularization method (which is most helpful in the small data setting), and a way to reduce oversmoothing to enable training very deep models (which is most helpful in the large data setting). It is, therefore, unclear how the performance of NN would vary with the size of a dataset. It would be good if the authors can shed some light on this.


**Summary Of The Paper:**

The paper presents a simple regularization method based on denoising, called Noisy Nodes (NN) for Graph Neural Networks to reduce oversmoothing. The NN method adds a small noise to the input node representation (and optionally edge representations) during training. An auxiliary loss is added for predicting the uncorrupted node representation. The paper presents experiments using this method for a few popular GNNs (GNS, GCN etc) on different datasets and show improved performance, especially for very deep GNNs.


**Summary Of The Review:**

The paper presents a simple new method for improving GNNs. The proposed method is based on existing ideas used in other domains, but their application to GNNs is novel. Through extensive experiments, this method was shown to benefit a variety of GNN architectures across different datasets and tasks. The simplicity and generality of this method is likely to have a big impact on the field of GNNs, which prompts me to lean towards accepting the paper.

---

> ### Author Response · Authors · 2021-11-15
> **Response with Additional Analysis & Experiments**
>
> Dear reviewer,
>
> Thank you for your helpful review and your positive score. We have conducted some additional experiments and made changes to the manuscript to address your concerns. We hope that these changes resolve your questions, and that you may consider raising your score.
>
> **“NN adds additional hyper parameters like noise standard deviation and weight of the auxiliary loss ....”**
>
> Thank you for this suggestion, we have conducted an ablation on noise for OGBG-MOLPCBA and provide it in the appendix in our paper. We repeat the results below, which show that our model gains benefits from using noise at multiple different levels, and seems to degrade slightly as we go above 0.05. In general we found trying a small set of values allowed us to find a good value.
>
> | Random Flip Probability | Mean AP           |
> |-------------------------|-------------------|
> | 0.01                   | 27.8% +- 0.002|
> | 0.03                    | 27.9% +- 0.003|
> | 0.05                    |28.1%+- 0.001 |
> | 0.1                     | 28.0% +- 0.003 |
> | 0.2                     | 27.7% +- 0.002 |
>
> We report our hyperparameters for the weighting coefficient in the appendix and find that they vary by dataset. For example, in OC20 the Noisy Nodes loss  and energy loss are equally weighted (weight of 1.), whereas for OGBG-PCQM4M and OGBG-MOLPCBA the weighting coefficient is 0.1. We found evaluating a small number (<5) of coefficients allowed us to find the optimal value. We have updated our paper to more clearly sign post this.
>
> **“In table 7, the 16 layer model is only shown with NN. Please add a column to show performance without NN.”**
>
> We are currently running this experiment and will update our paper when it has completed.
>
>
> **“While the current results are very impressive, some of the best models for molecular prediction...”**
>
> Thank you for raising this point, we agree this is an important question that we should address in the paper.
>
> In these models the spatial features of the molecule may be higher order terms such as distances and angles. In this case, you can adapt Noisy Nodes to predict pairwise interatomic distances or the angles of the ground truth molecular graph. We have added text in our manuscript to explain this point and outline how NN can be integrated with alternative architectures.
>
> **“The paper shows benefits of scaling the depth up to 50 layers. It would be good to show what happens beyond this depth....”**
>
> Thank you for raising this point, we agree that we could improve the paper to demonstrate the effects of scaling.
>
> In the case of our OC dataset, we have results on up to 100 layers (Table 1), and see that performance still improves but not as significantly. There is  a diminishing effect to depth and we have edited our paper to reflect this.
>
>
> **“NN can be seen as both a regularization method (which is most helpful in the small data setting), and a way to reduce oversmoothing to enable training very deep models ...”**
>
> Thank you for asking this question.
>
> OC20 is the largest of our 3D datasets, and where we find the most benefit from depth for 3D tasks at 100 layers. As the size of the dataset decreases, so does the optimal depth, and we find that QM9 is optimal at 30. The majority of the effect, however, can be seen at 3 layers (Figure 3) and so we speculate that the dominant effect is regularisation.
>
> For non-spatial tasks, we find that OGBG-PCQM4M is optimal with a depth of 50, whereas OGBG-MOLPCBA has diminishing returns after 16 layers. We have updated our manuscript to make the point that depth is most beneficial for large data.
>
> In addition to the above, you may find useful the following analysis useful which shows Noisy Nodes can combine with other ways to address oversmoothing. We now have a comparison with DropNode & DropEdge applied to OGB-MOLPCBA  with a 16 layer MPNN + Virtual Node in the Appendix. We repeat the results below.
>
>
> |                      | Mean AP              |
> |----------------------|----------------------|
> | MPNN | 27.5%±0.001 |
> | MPNN + Drop Node   | 27.5%±0.004|
> | MPNN + Drop Node  + Noisy Nodes | 28.2%±0.005|
>
>
> Above we see that DropNode does not improve the performance of our network, However, Drop Node combines favourably with Noisy Nodes.
>
> We find the same effect for DropEdge:
>
> |                      | Mean AP              |
> |----------------------|----------------------|
> | MPNN | 27.5%±0.002 |
> | MPNN + Drop Edge  | 27.5%±0.001 |
> | MPNN + Drop Edge + Noisy Nodes | 27.8%±0.002 |
>
> Interestingly, we find that DropEdge and DropNode do not combine well together, reaching a performance of 27.0%±0.003. However, Noisy Nodes combined with both methods reaches a Mean AP of 27.6% +- 0.0004
>
> Finally, in Figure 8 of the updated text we analyse oversmoothing in the node latents for DropNode & DropEdge. In this analysis we find that Noisy Nodes is significantly more successful at preserving per node latent diversity than either DropEdge or DropNode for MPNNs.

---

> ### Author Response · Authors · 2021-11-24
> **Please let us know if you have other questions or comments and we will be very happy to respond.**
>
> We believe we have addressed your concerns, but please let us know if you have other questions or comments and we will be very happy to respond.

---

> > ### Comment · Reviewer_yQtf · 2021-11-27
> > **Response**
> >
> > Thank you. I am happy with your response, and my concerns have been addressed.

---

### Official Review · Reviewer_kA11 · 2021-11-02

**Correctness:** 4
**Technical Novelty And Significance:** 2
**Empirical Novelty And Significance:** 3
**Recommendation:** 8
**Confidence:** 2

**Main Review:**

* Pros:

1. The node noise method is simple and easy-to-use with significant improvement on 3D molecular prediction tasks and non-spatail tasks, which may make it a standard trick in the future.
2. The abalation study is comprehensive.

* Concerns:

1. I agree that it would be interesting to incorporate the methods into small data regime.
2. Such an autoencoder structure may guide the learning process in GNNs into some bias. For example, it would be intriguing to see how to use it to analyze whether GNNs can keep focusing on local information after stacking many layers. (This question is a little bit derivating from the original motivation of this paper.)

**Summary Of The Paper:**

To overcome oversmoothing in GNNs, this paper proposes node noise with noise-correction training, similar to denoising autoencoder, to make node representation distinguishable. On 3D molecular property prediction tasks, it improves GNS significantly, achieving state-of-the-art. It also improves GCNs on non-spatial tasks.

**Summary Of The Review:**

I recommend to accept it for its straightforward intuition, simple methods and strong experiment results.

---

> ### Author Response · Authors · 2021-11-15
> **Additional New Analysis**
>
> Dear reviewer,
>
> Thank you for your helpful review and positive feedback.  We have conducted additional experiments and analysis to support our results which we hope will improve your confidence in your score.
>
>
> **“I agree that it would be interesting to incorporate the methods into small data regime.”**
>
> We agree, and we plan to address this in future work.
>
>
> **“Such an autoencoder structure may guide the learning process in GNNs into some bias. For example, it would be intriguing to see how to use it to analyze whether GNNs can keep focusing on local information after stacking many layers. (This question is a little bit derivating from the original motivation of this paper.)”**
>
> We agree that this would be a very interesting piece of analysis which we would like to consider in further work.
>
> You may also find interesting the following additional analysis and experiments we have conducted.
>
> We have conducted an ablation on noise for OGBG-MOLPCBA for a 16 layer MPNN + Noisy Nodes and provide it in the appendix of our paper. We repeat the results below, which show that our model gains benefits from using noise at multiple different levels, and seems to degrade slightly as we go above 0.05. In general we found trying a small set of values allowed us to find a good value.
>
> | Random Flip Probability | Mean AP           |
> |-------------------------|-------------------|
> | 0.01                   | 27.8% +- 0.002|
> | 0.03                    | 27.9% +- 0.003|
> | 0.05                    |28.1%+- 0.001 |
> | 0.1                     | 28.0% +- 0.003 |
> | 0.2                     | 27.7% +- 0.002 |
>
> In addition we show that Noisy Nodes can compose with alternatives ways to address oversmoothing. We have now included an ablation with DropNode & DropEdge applied to OGB-MOLPCBA  with a 16 layer MPNN + Virtual Node in the Appendix. We repeat the results here for ease of comparison.
>
>
> |                      | Mean AP              |
> |----------------------|----------------------|
> | MPNN | 27.5%±0.001 |
> | MPNN + Drop Node   | 27.5%±0.004|
> | MPNN + Drop Node  + Noisy Nodes | 28.2%±0.005|
>
>
> Above we see that DropNode does not improve the performance of our network, However, Drop Node combines favorably with Noisy Nodes.
>
> We find the same effect for DropEdge:
>
> |                      | Mean AP              |
> |----------------------|----------------------|
> | MPNN | 27.4%±0.002 |
> | MPNN + Drop Edge  | 27.5%±0.001 |
> | MPNN + Drop Edge + Noisy Nodes | 27.8%±0.002 |
>
> Interestingly, we find that DropEdge and DropNode do not combine well together, reaching a performance of 27.0%±0.003. However, Noisy Nodes combined with both methods reaches a Mean AP of 27.6% 0 0.0004
>
> Finally, in Figure 8 of the updated text we analyse oversmoothing in the node latents for each of these methods. In this analysis we find that Noisy Nodes is significantly more successful at preserving per node latent diversity than either DropEdge or DropNode for MPNNs.

---

> > ### Comment · Reviewer_kA11 · 2021-11-29
> > **Reviewer response**
> >
> > I would like to keep my score with relatively low confidence.

---

> ### Author Response · Authors · 2021-11-24
> **Please let us know if you have other questions or comments.**
>
> We believe we have addressed your concerns, but please let us know if you have other questions or comments and we will be very happy to respond.

---

### Decision · Program_Chairs · 2022-01-20

**Decision:**

Accept (Poster)

**Comment:**

This is a borderline paper. The most enthusiastic reviewer does not have much confidence in the score. The other reviewers think the paper has some value after the rebuttal, but also feel there is little technical novelty. The proposed applications of the approach are interesting.

After reading the reviews, rebuttal, and the paper, I agree that there is little technical novelty. The idea of adding node-label noise to a GNN to improve GNN expressiveness dates back to (Murphy et al., 2019) and has been also explored by (Dasoulas et al., 2019), (Vignac et al., 2020), (Loukas, 2020) among others [one of which is suggested by a reviewer] (this literature is entirely missing from the paper). The paper has some novelty in proposing a regularization method for tackling the node-level noise by augmenting the loss function with a denoising term. The oversmoothing justification is not properly investigated (whether the proposed solution really solves the issue in practice).

If there is space in the borderline decision boundary, this paper could be a worthwhile inclusion.

Dasoulas, G., Santos, L.D., Scaman, K. and Virmaux, A., 2019. Coloring graph neural networks for node disambiguation. arXiv preprint arXiv:1912.06058.
Loukas, A., 2020. How hard is to distinguish graphs with graph neural networks?. arXiv preprint arXiv:2005.06649.
Vignac, C., Loukas, A. and Frossard, P., 2020. Building powerful and equivariant graph neural networks with structural message-passing. arXiv preprint arXiv:2006.15107.
Murphy, R., Srinivasan, B., Rao, V. and Ribeiro, B., 2019, May. Relational pooling for graph representations. In International Conference on Machine Learning (pp. 4663-4673). PMLR.